# NON-DEEP NETWORKS

## ABSTRACT

Depth is the hallmark of deep neural networks. But more depth means more sequential computation and higher latency. This begs the question – is it possible to build high-performing "non-deep" neural networks? We show that it is. To do so, we use parallel subnetworks instead of stacking one layer after another. This helps effectively reduce depth while maintaining high performance. By utilizing parallel substructures, we show, for the first time, that a network with a depth of just 12 can achieve top-1 accuracy over $80\%$ on ImageNet, $96\%$ on CIFAR10, and $81\%$ on CIFAR100. We also show that a network with a low-depth (12) backbone can achieve an AP of $48\%$ on MS-COCO. We analyze the scaling rules for our design and show how to increase performance without changing the network's depth. Finally, we provide a proof of concept for how non-deep networks could be used to build low-latency recognition systems. We will open-source our code.

## 1 INTRODUCTION

Deep Neural Networks (DNNs) have revolutionized the fields of machine learning, computer vision, and natural language processing. As their name suggests, a key characteristic of DNNs is that they are deep. That is, they have a large depth, which can be defined as the length of the longest path from an input neuron to an output neuron. Often a neural network can be described as a linear sequence of layers, i.e. groups of neurons with no intra-group connections. In such cases, the depth of a network is its number of layers.

It has been generally accepted that large depth is an essential component for high-performing networks because depth increases the representational ability of a network and helps learn increasingly abstract features (He et al., 2016a). In fact, one of the primary reasons given for the success of ResNets is that they allow training very deep networks with as many as 1000 layers (He et al., 2016a). As such, state-of-the-art performance is increasingly achieved by training models with large depth, and what qualifies as "deep" has shifted from "2 or more layers" in the early days of deep learning to the "tens or hundreds of layers" routinely used in today's models. For example, as shown in Figure 1, competitive benchmarks such as ImageNet are dominated by very deep models (He et al., 2016a; Huang et al., 2017; Tan & Le, 2019) with at least 30 layers, whereas models with fewer than 30 layers perform substantially worse. The best-performing model with fewer than 20 layers has a top-1 accuracy of only 75.2, substantially lower than accuracies achievable with 30 or more layers when evaluated with a single image crop (He et al., 2015; Tan & Le, 2019).

But is large depth always necessary? This question is worth asking because large depth is not without drawbacks. A deeper network leads to more sequential processing and higher latency; it is harder to parallelize and less suitable for applications that require fast responses.

In this paper, we study whether it is possible to achieve high performance with "non-deep" neural networks, especially networks with $\sim 10$ layers. We find that, contrary to conventional wisdom, this is indeed possible. We present a network design that is non-deep and performs competitively against its deep counterparts. We refer to our architecture as ParNet (Parallel Networks). We show, for the first time, that *a classification network with a depth of just 12 can achieve accuracy greater than 80% on ImageNet, 96% on CIFAR10, and 81% on CIFAR100.* We also show that *a detection network with a low-depth (12) backbone can achieve an AP of $48\%$ on MS-COCO.* Note that the number of parameters in ParNet is comparable to state-of-the-art models, as illustrated in Figure 1.

A key design choice in ParNet is the use of parallel subnetworks. Instead of arranging layers sequentially, we arrange layers in parallel subnetworks. This design is "embarrassingly parallel",

in the sense that there are no connections between the subnetworks except at the beginning and the end. This allows us to reduce the depth of the network while maintaining high accuracy. It is worth noting that our parallel structures are distinct from "widening" a network by increasing the number of neurons in a layer.

ParNet not only helps us answer a scientific question about the necessity of large depth, but also offers practical advantages. Due to the parallel substructures, ParNet can be efficiently parallelized across multiple processors. We find that ParNet can be effectively parallelized and *outperforms ResNets in terms of both speed and accuracy*. Note that this is achieved despite the extra latency introduced by the communication between processing units. This shows that in the future, with possibly specialized hardware to further mitigate communication latency, ParNet-like architectures could be used for creating extremely fast recognition systems.

We also study the scaling rules for ParNet. Specifically, we show that ParNet can be effectively scaled by increasing width, resolution, and

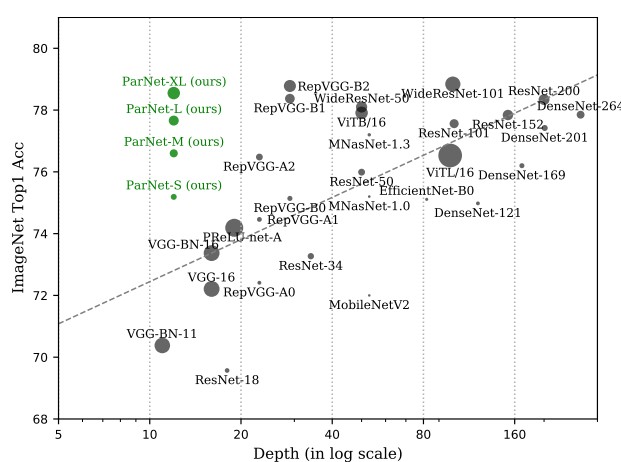

Figure 1: Top-1 accuracy on ImageNet vs. depth (in log scale) of various models. ParNet performs competitively to deep state-of-the-art neural networks while having much lower depth. Performance of prior models is as reported in the literature. Size of the circle is proportional to the number of parameters. Models are evaluated using a single 224×224 crop, except for ViTB-16 and ViTB-32 (Dosovitskiy et al., 2021), which fine-tunes at 384×384 and PReLU-net (He et al., 2015), which evaluates at 256×256. Models are trained for 90 to 120 epochs, except for parameter-efficient models such as MNASNet (Tan et al., 2019), MobileNet (Sandler et al., 2018), and EfficientNet (Tan & Le, 2019), which are trained for more than 300 epochs. For fairness, we exclude results with longer training, higher resolution, or multi-crop testing.

number of branches, all while keeping depth constant. We observe that the performance of ParNet does not saturate and increases as we increase computational throughput. This suggests that by increasing compute further, one can achieve even higher performance while maintaining small depth (~10) and low latency.

To summarize, our contributions are three-fold:

- We show, for the first time, that a neural network with a depth of only 12 can achieve high performance on very competitive benchmarks (80.7% on ImageNet, 96% on CIFAR10, 81% on CIFAR100).
- We show how parallel structures in ParNet can be utilized for fast, low-latency inference.
- We study the scaling rules for ParNet and demonstrate effective scaling with constant low depth.

## 2 RELATED WORK

**Analyzing importance of depth.** There exists a rich literature analyzing the importance of depth in neural networks. The classic work of Cybenko et al. showed that a single-layer neural network with sigmoid activations can approximate any function with arbitrarily small error (Cybenko, 1989). However, one needs to use a network with sufficiently large width, which can drastically increase the parameter count. Subsequent works have shown that, to approximate a function, a deep network with non-linearity needs exponentially fewer parameters than its shallow counterpart (Liang & Srikant, 2017). This is often cited as one of the major advantages of large depth.

Several works have also empirically analyzed the importance of depth and came to the conclusion that under a fixed parameter budget, deeper networks perform better than their shallow counter-

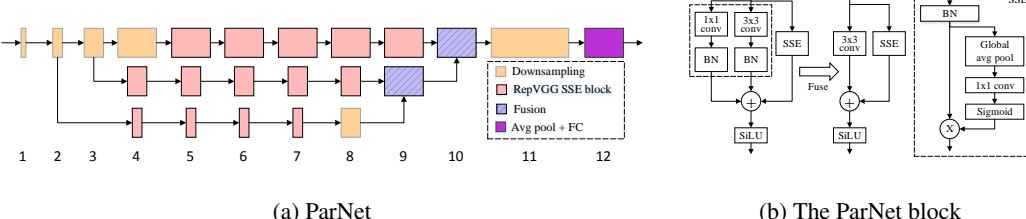

(a) ParNet            (b) The ParNet block

Figure 2: Schematic representation of ParNet and the ParNet block. ParNet has depth 12 and is composed of parallel substructures. The width of each block in (a) is proportional to the number of output channels in ParNet-M and the height reflects output resolution. The ParNet block consists of three parallel branches: 1×1 convolution, 3×3 convolution and Skip-Squeeze-and-Excitation (SSE). Once the training is done, the 1×1 and 3×3 convolutions can be fused together for faster inference. The SSE branch increases receptive field while not affecting depth.

parts (Eigen et al., 2013; Urban et al., 2017). However, in such analysis, prior works have only studied shallow networks with a linear, sequential structure, and it is unclear whether the conclusion still holds with alternative designs. In this work, we show that, contrary to conventional wisdom, a shallow network can perform surprisingly well, and the key is to have parallel substructures.

**Scaling DNNs.** There have been many exciting works that have studied the problem of scaling neural networks. Tan & Le (2019) showed that increasing depth, width, and resolution leads to effective scaling of convolutional networks. We also study scaling rules but focus on the low-depth regime. We find that one can increase the number of branches, width, and resolution to effectively scale ParNet while keeping depth constant and low. Zagoruyko & Komodakis (2016) showed that shallower networks with a large width can achieve similar performance to deeper ResNets. We also scale our networks by increasing their width. However, we consider networks that are much shallower – a depth of just 12 compared to 50 considered for ImageNet by Zagoruyko & Komodakis (2016) – and introduce parallel substructures.

**Shallow networks.** Shallow networks have attracted attention in theoretical machine learning. With infinite width, a single-layer neural network behaves like a Gaussian Process, and one can understand the training procedure in terms of kernel methods (Jacot et al., 2018). However, such models do not perform competitively when compared to state-of-the-art networks (Li et al., 2019). We provide empirical proof that non-deep networks can be competitive with their deep counterparts.

**Multi-stream networks.** Multi-stream neural networks have been used in a variety of computer vision tasks such as segmentation (Chen et al., 2016; 2017), detection (Lin et al., 2017), and video classification (Wu et al., 2016). The HRNet architecture maintains multi-resolution streams throughout the forward pass (Wang et al., 2020); these streams are fused together at regular intervals to exchange information. We also use streams with different resolutions, but our network is much shallower (12 vs. 38 for the smallest HRNet for classification) and the streams are fused only once, at the very end, making parallelization easier.

## 3 METHOD

In this section, we develop and analyze ParNet, a network architecture that is much less deep but still achieves high performance on competitive benchmarks. ParNet consists of parallel substructures that process features at different resolutions. We refer to these parallel substructures as *streams*. Features from different streams are fused at a later stage in the network, and these fused features are used for the downstream task. Figure 2a provides a schematic representation of ParNet.

### 3.1 PARNET BLOCK

In ParNet, we utilize VGG-style blocks (Simonyan & Zisserman, 2015). To see whether non-deep networks can achieve high performance, we empirically find that VGG-style blocks are more suitable than ResNet-style blocks (Table 8). In general, training VGG-style networks is more difficult than their ResNet counterparts (He et al., 2016a). But recent work shows that it is easier to train net-

works with such blocks if one uses a "structural reparameterization" technique (Ding et al., 2021). During training, one uses multiple branches over the 3×3 convolution blocks. Once trained, the multiple branches can be fused into one 3×3 convolution. Hence, one ends up with a plain network consisting of only 3×3 block and non-linearity. This reparameterization or fusion of blocks helps reduce latency during inference.

We borrow our initial block design from Rep-VGG (Ding et al., 2021) and modify it to make it more suitable for our non-deep architecture. One challenge with a non-deep network with only 3×3 convolutions is that the receptive field is rather limited. To address this, we build a Skip-Squeeze-Excitation (SSE) layer which is based on the Squeeze-and-Excitation (SE) design (Hu et al., 2018). Vanilla Squeeze-and-Excitation is not suitable for our purpose as it increases the depth of the network. Hence we use a Skip-Squeeze-Excitation design which is applied alongside the skip connection and uses a single fully-connected layer. We find that this design helps increase performance (Table 7). Figure 2b provides a schematic representation of our modified Rep-VGG block with the Skip-Squeeze-Excitation module. We refer to this block as the *RepVGG-SSE*.

One concern, especially with large-scale datasets such as ImageNet, is that a non-deep network may not have sufficient non-linearity, limiting its representational power. Thus we replace the ReLU activation with SiLU (Ramachandran et al., 2017).

## 3.2 Downsampling and Fusion Block

Apart from the RepVGG-SSE block, whose input and output have the same size, ParNet also contains *Downsampling* and *Fusion* blocks. The Downsampling block reduces resolution and increases width to enable multi-scale processing, while the Fusion block combines information from multiple resolutions.

In the Downsampling block, there is no skip connection; instead, we add a single-layered SE module parallel to the convolution layer. Additionally, we add 2D average pooling in the 1×1 convolution branch. The Fusion block is similar to the Downsampling block but contains an extra concatenation layer. Because of concatenation, the input to the Fusion block has twice as many channels as a Downsampling block. Hence, to reduce the parameter count, we use convolution with group 2. Please refer to Figure A1 in the appendix for a schematic representation of the Downsampling and Fusion blocks.

## 3.3 Network Architecture

Figure 2a shows a schematic representation of the ParNet model that is used for the ImageNet dataset. The initial layers consist of a sequence of Downsampling blocks. The outputs of Downsampling blocks 2, 3, and 4 are fed respectively to streams 1, 2, and 3. We empirically find 3 to be the optimal number of streams for a given parameter budget (Table 10). Each stream consists of a series of RepVGG-SSE blocks that process the features at different resolutions. The features from different streams are then fused by Fusion blocks using concatenation. Finally, the output is passed to a Downsampling block at depth 11. Similar to RepVGG (Ding et al., 2021), we use a larger width for the last downsampling layer. Table A1 in the appendix provides a complete specification of the scaled ParNet models that are used in ImageNet experiments.

In CIFAR the images are of lower resolution, and the network architecture is slightly different from the one for ImageNet. First, we replace the Downsampling blocks at depths 1 and 2 with RepVGG-SSE blocks. To reduce the number of parameters in the last layer, we replace the last Downsampling block, which has a large width, with a narrower 1×1 convolution layer. Also, we reduce the number of parameters by removing one block from each stream and adding a block at depth 3.

## 3.4 Scaling ParNet

With neural networks, it is observed that one can achieve higher accuracy by scaling up network size. Prior works (Tan & Le, 2019) have scaled width, resolution, and depth. Since our objective is to evaluate whether high performance can be achieved with small depth, we keep the depth constant and instead scale up ParNet by increasing width, resolution, and the number of streams.

Table 1: Depth vs. performance on ImageNet. We test on images with $224 \times 224$ resolution. We rerun ResNets (He et al., 2016a) in our training regime for fairness, thus boosting their accuracy. Our ParNet models perform competitively with ResNets while having a low constant depth.

| Model | Depth (in M) | Top-1 Acc. | Top-5 Acc. |
|---|---|---|---|
| ResNet | 18 | 69.57 | 89.24 |
| ResNet | 34 | 73.27 | 91.26 |
| ResNet-Bottleneck | 50 | 75.99 | 92.98 |
| ResNet-Bottleneck | 101 | 77.56 | 93.79 |
| ResNet (ours) | 18 | 70.15 | 89.55 |
| ResNet (ours) | 34 | 74.12 | 91.89 |
| ResNet-Bottleneck (ours) | 26 | 77.53 | 93.87 |
| ResNet-Bottleneck (ours) | 101 | 79.63 | 94.68 |
| ParNet-S | 12 | 75.19 | 92.29 |
| ParNet-M | 12 | 76.60 | 93.02 |
| ParNet-L | 12 | 77.66 | 93.6 |
| ParNet-XL | 12 | 78.55 | 94.13 |

Table 2: Speed and performance of ParNet vs. ResNet. Because of parallel substructures, ParNet can be distributed across multiple GPUs. In spite of communication overhead, ParNet is faster than similar-performing ResNets.

| Model | Depth | Top-1 Acc. | Speed (samples/s) | # Param (in M) | Flops (in B) |
|---|---|---|---|---|---|
| ParNet-L | 12 | **77.66** | **249** | 54.9 | 26.7 |
| ResNet34 | 34 | 74.12 | 248 | 21.8 | 7.3 |
| ResNet50 | 50 | 77.53 | 207 | 25.6 | 8.2 |
| ParNet-XL | 12 | **78.55** | **230** | 85.0 | 41.5 |
| ResNet50 | 50 | 77.53 | 207 | 25.6 | 8.2 |

Table 3: Fusing features and parallelizing the substructures across GPUs improves the speed of ParNet. Speed was measured on a GeForce RTX 3090 with Pytorch 1.8.1 and CUDA 11.1.

| Model | Top-1 Acc. | Speed (samples/sec) | Latency (ms) |
|---|---|---|---|
| ParNet-L (Unfused) | 77.66 | 112 | 8.95 |
| ParNet-L (Fused, Single GPU) | 77.66 | 154 | 6.50 |
| ParNet-L | 77.66 | **249** | **4.01** |

For CIFAR10 and CIFAR100, we increase the width of the network while keeping the resolution at 32 and the number of streams at 3. For ImageNet, we conduct experiments by varying all three dimensions (Figure 3).

## 3.5 Practical Advantages of Parallel Architectures

The current process of 5 nm lithography is approaching the 0.5 nm size of the silicon crystal lattice, and there is limited room to further increase processor frequency. This means that faster inference of neural networks must come from parallelization of computation. The growth in the performance of single monolithic GPUs is also slowing down (Arunkumar et al., 2017). The maximum die size achievable with conventional photolithography is expected to plateau at $\sim$800mm$^2$ (Arunkumar et al., 2017). On the whole, a plateau is expected not only in processor frequency but also in the die size and the number of transistors per processor.

To solve this problem, there are suggestions for partitioning a GPU into separate basic modules that lie in one package. These basic modules are easier to manufacture than a single monolithic GPU on a large die. Large dies have a large number of manufacturing faults, resulting in low yields (Kannan et al., 2015). Recent work has proposed a Multi-Chip-Module GPU (MCM-GPU) on an interposer, which is faster than the largest implementable monolithic GPU. Replacing large dies with medium-size dies is expected to result in lower silicon costs, significantly higher silicon yields, and cost advantages (Arunkumar et al., 2017).

Even if several chips are combined into a single package and are located on one interposer, the data transfer rate between them will be less than the data transfer rate inside one chip, because the lithography size of the chip is smaller than the lithography size of the interposer. Such chip designs thus favor partitioned algorithms with parallel branches that exchange limited data and can be executed independently for as long as possible. All these factors make non-deep parallel structures advantageous in achieving fast inference, especially with tomorrow's hardware.

## 4 Results

**Experiments on ImageNet.** ImageNet (Deng et al., 2009) is a large-scale image classification dataset with high-resolution images. We evaluate on the ILSVRC2012 (Russakovsky et al., 2015) dataset, which consists of 1.28M training images and 50K validation images with 1000 classes. We train our models for 120 epochs using the SGD optimizer, a step scheduler with a warmup for first

Table 4: Non-deep networks can be used as backbones for fast and accurate object detection systems. Speed is measured on a single RTX 3090 using Pytorch 1.8.1 and CUDA 11.1.

| Model | Backbone Depth | MS-COCO AP | Latency (in ms) |
|---|---|---|---|
| YOLOv4-CSP (official) | 64 | 46.2 | 21.0 |
| YOLOv4-CSP (Ours) | 64 | 47.6 | 20.0 |
| ParNet-XL (Ours) | 12 | 47.5 | 18.7 |
| ParNet-XL-CSP (Ours) | 12 | **48.0** | **16.4** |

Table 5: A network with depth 12 can get 80.72% top-1 accuracy on ImageNet. We show how various strategies can be used to boost the performance of ParNet.

| Model | Top-1 Acc. | Top-5 Acc. |
|---|---|---|
| ParNet-XL | 78.55 | 94.13 |
| + Longer Training | 78.97 | 94.51 |
| + Train & Test Res. 320 | 80.32 | 94.95 |
| + 10-crop testing | 80.72 | 95.38 |

5 epochs, a learning rate decay of 0.1 at every $30^{th}$ epoch, an initial learning rate of 0.8, and a batch size of 2048 (256 per GPU). If the network does not fit in memory, we decrease the batch size and the learning rate proportionally, for example, a decrease to a learning rate of 0.4 and a batch size of 1024. Unless otherwise specified, the network is trained at 224×224 resolution. We train our networks with the cross-entropy loss with smoothed labels (Szegedy et al., 2016). We use cropping, flipping, color-jitter, and rand-augment (Cubuk et al., 2020) data augmentations.

In Table 1, we show the performance of ParNet on ImageNet. We find that one can achieve surprisingly high performance with a depth of just 12. For a fair comparison with ResNets, we retrain them with our training protocol and data augmentation, which improves the performance of ResNets over the official number. Notably, we find that ParNet-S outperform ResNet34 by over 1 percentage point with a lower parameter count (19M vs. 22M). ParNet also achieves comparable performance to ResNet with the bottleneck design, while having 4 to 8 times less depth. For example, ParNet-L performs as well as ResNet50 and gets a top-1 accuracy of 77.66 as compared to 77.53 achieved by ResNet50. Similarly, ParNet-XL performs comparably to ResNet101 and gets a top-5 accuracy of 94.13, in comparison to 94.68 achieved by ResNet101, while being 8 times shallower.

In Table 2, we find that ParNet performs favourably to ResNet when comparing accuracy and speed, however with more parameters and flops. For example, ParNet-L achieves faster speed and better accuracy than ResNet34 and ResNet50. Similarly, ParNet-XL achieves faster speed and better accuracy than ResNet50, however with more parameters and flops. This suggests that depending on the use case, one can use ParNets instead of ResNets to trade off speed with more parameters and flops. Note that the high speed can be achieved by leveraging the parallel sub-structures that could be distributed across GPUs (more details below).

In Table 3, we test speed for three variants of ParNet: unfused, fused, and multi-GPU. The unfused variant consists of 3×3 and 1×1 branches in the RepVGG-SSE block. In the fused variant, we use the structural-reparametrization trick to merge the 3×3 and 1×1 branches into a single 3×3 branch (Section 3.1). For both fused and unfused versions, we use a single GPU for inference, while for the multi-GPU version, we use 3 GPUs. For the multi-GPU version, each stream is launched on a separate GPU. When all layers in a stream are processed, the results from two adjacent streams are concatenated on one of the GPUs and processed further. For transferring data across GPUs we use the NCCL backend in Pytorch.

We find that ParNet can be effectively parallelized across GPUs for fast inference. This is achieved in spite of the communication overhead. With specialized hardware for reducing communication latency, even faster speeds could be achieved.

**Boosting Performance.** Table 5 demonstrates additional ways of increasing the performance of ParNet, such as using higher-resolution images, a longer training regime (200 epochs, cosine annealing), and 10-crop testing. This study is useful in assessing the accuracy that can be achieved by non-deep models on large-scale datasets like ImageNet. By employing various strategies we can boost the performance of ParNet-XL from 78.55 to 80.72. Notably, we reach a top-5 accuracy of 95.38, which is higher than the oft-cited human performance level on ImageNet (Russakovsky et al., 2015). Although this does not mean that machine vision has surpassed human vision, it provides a sense of how well ParNet performs. To the best of our knowledge, this is the first instance of such "human-level" performance achieved by a network with a depth of just 12.

Table 6: Performance of various architectures on CIFAR10 and CIFAR100. Similar-sized models are grouped together. ParNet performs competitively with deep state-of-the-art architectures while having a much smaller depth. Best performance is **bolded**. The second and third best performing model in each model size block are underlined.

| Architecture | Depth | Params (in Millions) | CIFAR10 Error | CIFAR100 Error |
|---|---|---|---|---|
| ResNet110 (official) | 110 | 1.7 | 6.61 | – |
| ResNet110 (reported in Huang et al. (2016)) | 110 | 1.7 | 6.41 | 27.22 |
| ResNet (Stochastic Depth) (Huang et al., 2016) | 110 | 1.7 | 5.23 | 24.58 |
| ResNet (pre-act) (He et al., 2016b) | 164 | 1.7 | 5.46 | 24.33 |
| DenseNet (Huang et al., 2017) | 40 | 1.0 | 5.24 | 24.42 |
| DenseNet (Bottleneck+Compression) (Huang et al., 2017) | 100 | 0.8 | **4.51** | **22.27** |
| ParNet (Ours) | 12 | 1.3 | 5.0 | 24.62 |
| ResNet (Stochastic Depth) (Huang et al., 2016) | 1202 | 10.2 | 4.91 | – |
| ResNet (pre-act) (He et al., 2016b) | 1001 | 10.2 | 4.62 | 22.71 |
| WideResNet (Zagoruyko & Komodakis, 2016) | 40 | 8.9 | 4.53 | 21.18 |
| WideResNet (Zagoruyko & Komodakis, 2016) | 16 | 11.0 | 4.27 | 20.43 |
| WideResNet (SE) (Hu et al., 2018) | 32 | 12.0 | 3.88 | 19.14 |
| FractalNet (Compressed) (Larsson et al., 2017) | 41 | 22.9 | 5.21 | 21.49 |
| DenseNet (Huang et al., 2017) | 100 | 7.0 | 4.10 | 20.20 |
| DenseNet (Bottleneck+Compression) (Huang et al., 2017) | 250 | 15.3 | **3.62** | **17.60** |
| ParNet (Ours) | 12 | 15.5 | 3.90 | 20.02 |
| WideResNet (Zagoruyko & Komodakis, 2016) | 28 | 36.5 | 4.00 | 19.25 |
| WideResNet (Dropout in Res-Block) (Zagoruyko & Komodakis, 2016) | 28 | 36.5 | 3.89 | 18.85 |
| FractalNet (Larsson et al., 2017) | 21 | 36.8 | 5.11 | 22.85 |
| FractalNet (Dropout+Drop-path) (Larsson et al., 2017) | 21 | 36.8 | 4.59 | 23.36 |
| DenseNet (Huang et al., 2017) | 100 | 27.2 | 3.74 | 19.25 |
| DenseNet (Bottleneck+Compression) (Huang et al., 2017) | 190 | 25.6 | **3.46** | **17.18** |
| ParNet (Ours) | 12 | 35 | 3.88 | 18.65 |

**Experiments on MS-COCO.** MS-COCO (Lin et al., 2014) is an object detection dataset which contains images of everyday scenes with common objects. We evaluate on the COCO-2017 dataset, which consists of 118K training images and 5K validation images with 80 classes.

To test whether a non-deep network such as ParNet can work for object detection, we replace the backbone of state-of-the-art single stage detectors with ParNet. Specifically, we replace the CSP-Darknet53s backbone from YOLOv4-CSP (Wang et al., 2021a) with ParNet-XL, which is much shallower (64 vs. 12). We use the head and reduced neck from the YOLOR-D6 model, and train and test these models using the official YOLOR code (Wang et al., 2021b). We retrain YOLOv4-CSP (Wang et al., 2021a) with our protocol (same neck, same head, same training setup) for fair comparison and it improves performance over the official model. Additionally, for fair comparison, we test the ParNet-XL-CSP model by applying the CSP (Wang et al., 2021a) approach to ParNet-XL. We find that ParNet-XL and ParNet-XL-CSP are faster than the baseline even at higher image resolution. We thus use higher image resolution for ParNet-XL and ParNet-XL-CSP.

In Table 4 we find that even on a single GPU, ParNet achieves higher speed than strong baselines. This demonstrates how non-deep networks could be used to make fast object detection systems.

**Experiments on CIFAR.** The CIFAR datasets consist of colored natural images with $32\times32$ pixels. CIFAR-10 consists of images drawn from 10 and CIFAR-100 from 100 classes. The training and test sets contain 50,000 and 10,000 images respectively. We adopt a standard data augmentation scheme (mirroring/shifting) that is widely used for these two datasets (He et al., 2016a; Zagoruyko & Komodakis, 2016; Huang et al., 2017). We train for 400 epochs with a batch size of 128. The initial learning rate is 0.1 and is decreased by a factor of 5 at 30%, 60%, and 80% of the epochs as in (Zagoruyko & Komodakis, 2016). Similar to prior works (Zagoruyko & Komodakis, 2016; Huang et al., 2016), we use a weight decay of 0.0003 and set dropout in the convolution layer at 0.2 and dropout in the final fully-connected layer at 0.2 for all our networks on both datasets. We train each network on 4 GPUs (a batch size of 32 per GPU) and report the final test set accuracy.

Table 6 summarizes the performance of various networks on CIFAR10 and CIFAR100. We find that ParNet performs competitively with state-of-the-art deep networks like ResNets and DenseNets while using a much lower depth and a comparable number of parameters. ParNet outperforms

Table 7: Ablation of various choices for ParNet. Data augmentation, SiLU activation, and Skip-Squeeze-Excitation (SSE) improve performance.

| Model | Params (in M) | Top-1 Acc. | Top-5 Acc. |
|---|---|---|---|
| Baseline | 32.4 | 75.02 | 92.15 |
| + Data Augmentation | 32.4 | 75.12 | 92.00 |
| + SSE | 35.6 | 76.55 | 93.01 |
| + SiLU | 35.6 | 76.60 | 93.07 |

Table 8: ParNet outperforms non-deep ResNet variants. At depth 12, VGG-style blocks outperform ResNet blocks, and three branches outperform a single branch.

| Name | Block | Branch | Depth | Params (in M) | Top-1 Acc. |
|---|---|---|---|---|---|
| ResNet12-Wide | ResNet | 1 | 12 | 39.0 | 73.52 |
| ResNet14-Wide-BN | ResNet-BN | 1 | 14 | 39.0 | 72.06 |
| ResNet12-Wide-BN | ResNet-SSE | 1 | 12 | 39.0 | 73.91 |
| ParNet-M-OneStream | RepVGG-SSE | 1 | 12 | 36.0 | 75.83 |
| ParNet-M | RepVGG-SSE | 3 | 12 | 35.9 | 76.6 |

Table 9: ParNet outperforms ensembles across different parameter budgets.

| | ParNet-M | Single Stream | ParNet-L | Ensemble (2 Single Streams) | ParNet-XL | Ensemble (3 Single Streams) |
|---|---|---|---|---|---|---|
| # Param | 35.9 | 36 | 55 | 72 | 85.3 | 108 |
| ImageNet Top-1 | **76.60** | 75.83 | **77.66** | 77.20 | **78.55** | 77.68 |

ResNets that are 10 times deeper on CIFAR10 (5.0 vs. 5.23) while using a lower number of parameters (1.3M vs. 1.7M). Similarly, ParNet outperforms ResNets that are 100 times deeper on CIFAR10 (3.90 vs. 4.62) and CIFAR100 (20.02 vs. 22.71) while using 50% more parameters (15.5M vs. 10.2M).

ParNet performs as well as vanilla DenseNet models (Huang et al., 2017) with comparable parameter counts while using 3-8 times less depth. For example, on CIFAR100, ParNet (depth 12) achieves an error of 18.65 with 35M parameters and DenseNet (depth 100) achieves an error of 19.25 with 27.2M parameters. ParNet also performs on par or better than Wide ResNets (Zagoruyko & Komodakis, 2016) while having 2.5 times less depth. The best performance on the CIFAR dataset under a given parameter count is achieved by DenseNets with the bottleneck and reduced width (compression) design, although with an order of magnitude larger depth than ParNet.

Overall, it is surprising that a mere depth-12 network could achieve an accuracy of 96.12% on CIFAR10 and 81.35% on CIFAR100. This further indicates that non-deep networks can work as well as deeper counterparts.

**Ablation Studies.** To test if we can trivially reduce the depth of ResNets and make them wide, we test three ResNet variants: ResNet12-Wide, ResNet14-Wide-BN, and ResNet12-Wide-SSE. ResNet12-Wide uses the basic ResNet block and has depth 12, while ResNet14-Wide-BN uses the bottleneck ResNet block and has depth 14. Note that with the bottleneck ResNet block, one cannot achieve a depth lower than 14 while keeping the original ResNet structure as there has to be 1 initial convolution layer, 4 downsampling blocks ($3 \times 4 = 12$ depth), and 1 fully-connected layer. We find that ResNet12-Wide outperforms ResNet14-Wide-BN with the same parameter count. We additionally add SSE block and SiLU activation to ResNet12-Wide to create ResNet12-Wide-SSE to further control for confounding factors. We find that ParNet-M outperforms all the ResNet variants which have depth 12 by 2.7 percentage points, suggesting that trivially reducing depth and increasing width is not as effective as our approach. We show that the model with three branches performs better than a model with a single branch. We also show that with everything else being equal, using VGG-style blocks leads to better performance than the corresponding ResNet architecture. Architectural choices in ParNet like parallel substructures and VGG-style blocks are crucial for high performance at lower depths.

Table 7 reports ablation studies over various design choices for our network architecture and training protocol. We show that each of the three decisions (rand-augment data augmentation, SiLU activation, SSE block) leads to higher performance. Using all three leads to the best performance.

In Table 10, we evaluate networks with the same total number of parameters but with different numbers of branches: 1, 2, 3, and 4. We show that for a fixed number of parameters, a network with 3 branches has the highest accuracy and is optimal in both cases, with a network resolution of 224x224 and 320x320.

Table 10: Performance vs. number of streams. For a fixed parameter budget, 3 streams is optimal.

| | # of Branches (Res. 224) | | | | # of Branches (Res. 320) | | | |
| | 1 | 2 | 3 | 4 | 1 | 2 | 3 | 4 |
|---|---|---|---|---|---|---|---|---|
| ImageNet Top-1 | 75.83 | 76.1 | **76.75** | 76.34 | 76.91 | 77.12 | **77.56** | 77.46 |

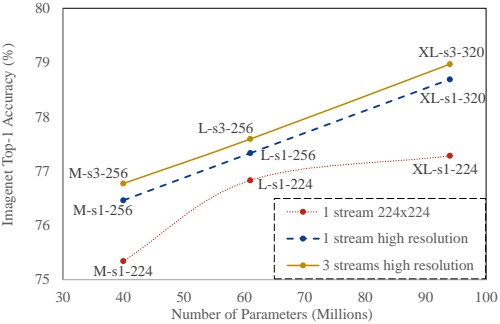 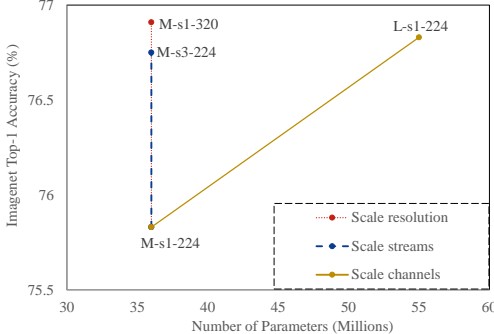

Figure 3: Performance of ParNet increases as we increase the number of streams, input resolution, and width of convolution, while keeping depth fixed. The left plot shows that under a fixed parameter count, the most effective way to scale ParNet is to use 3 branches and high resolution. The right plot shows the impact on performance by changing only one of the aforementioned factors. We do not observe saturation in performance, indicating that ParNets could be scaled further to increase performance while maintaining low depth.

**ParNet vs. Ensembles.** Another approach to network parallelization is the creation of ensembles consisting of multiple networks. Therefore, we compare ParNet to ensembles. In Table 9, we find that ParNet outperforms ensembles while using fewer parameters.

**Scaling ParNet.** Neural networks can be scaled by increasing resolution, width, and depth (Tan & Le, 2019). Since we are interested in exploring the performance limits of constant-depth networks, we scale ParNet by varying resolution, width, and the number of streams. Figure 3 shows that each of these factors increases the accuracy of the network. Also, we find that increasing the number of streams is more cost-effective than increasing the number of channels in terms of accuracy versus parameter count. Further, we find that the most effective way to scale ParNet is to increase all three factors simultaneously. Because of computation constraints, we could not increase the number of streams beyond 3, but this is not a hard limitation. Based on these charts, we see no saturation in performance while scaling ParNets. This indicates that by increasing compute, one could achieve even higher performance with ParNet while maintaining low depth.

## 5 DISCUSSION

We have provided the first empirical proof that non-deep networks can perform competitively with their deep counterparts in large-scale visual recognition benchmarks. We showed that parallel substructures can be used to create non-deep networks that perform surprisingly well. We also demonstrated ways to scale up and improve the performance of such networks without increasing depth.

Our work shows that there exist alternative designs where highly accurate neural networks need not be deep. Such designs can better meet the requirements of future multi-chip processors. We hope that our work can facilitate the development of neural networks that are both highly accurate and extremely fast.

**Reproducibility Statement.** We will open-source our code. This would enable readers to reproduce our work.

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

# A  APPENDIX

| layer name | layer depth | output size | Small | | Medium | | Large | | Extra large | |
|---|---|---|---|---|---|---|---|---|---|---|
| Downsampling | 1 | W/2 × H/2 | 1×1, 64
3×3, 64
stride 2 | ×1 | 1×1, 64
3×3, 64
stride 2 | ×1 | 1×1, 64
3×3, 64
stride 2 | ×1 | 1×1, 64
3×3, 64
stride 2 | ×1 |
| Downsampling | 2 | W/4 × H/4 | 1×1, 96
3×3, 96
stride 2 | ×1 | 1×1, 128
3×3, 128
stride 2 | ×1 | 1×1, 160
3×3, 160
stride 2 | ×1 | 1×1, 200
3×3, 200
stride 2 | ×1 |
| Downsampling | 3 | W/8 × H/8 | 1×1, 192
3×3, 192
stride 2 | ×1 | 1×1, 256
3×3, 256
stride 2 | ×1 | 1×1, 320
3×3, 320
stride 2 | ×1 | 1×1, 400
3×3, 400
stride 2 | ×1 |
| Downsampling | 4 | W/16 × H/16 | 1×1, 384
3×3, 384
stride 2 | ×1 | 1×1, 512
3×3, 512
stride 2 | ×1 | 1×1, 640
3×3, 640
stride 2 | ×1 | 1×1, 800
3×3, 800
stride 2 | ×1 |
| Stream1 | 3-6 | W/4 × H/4 | 1×1, 96
3×3, 96
SSE | ×4 | 1×1, 128
3×3, 128
SSE | ×4 | 1×1, 160
3×3, 160
SSE | ×4 | 1×1, 200
3×3, 200
SSE | ×4 |
| Stream1-Downsampling | 8 | W/8 × H/8 | 1×1, 192
3×3, 192
stride 2 | ×1 | 1×1, 256
3×3, 256
stride 2 | ×1 | 1×1, 320
3×3, 320
stride 2 | ×1 | 1×1, 400
3×3, 400
stride 2 | ×1 |
| Stream2 | 4-8 | W/8 × H/8 | 1×1, 192
3×3, 192
SSE | ×5 | 1×1, 256
3×3, 256
SSE | ×5 | 1×1, 320
3×3, 320
SSE | ×5 | 1×1, 400
3×3, 400
SSE | ×5 |
| Stream2-Fusion | 9 | W/16 × H/16 | 1×1, 384
3×3, 384
stride 2 | ×1 | 1×1, 512
3×3, 512
stride 2 | ×1 | 1×1, 640
3×3, 640
stride 2 | ×1 | 1×1, 800
3×3, 800
stride 2 | ×1 |
| Stream3 | 5-9 | W/16 × H/16 | 1×1, 384
3×3, 384
SSE | ×5 | 1×1, 512
3×3, 512
SSE | ×5 | 1×1, 640
3×3, 640
SSE | ×5 | 1×1, 800
3×3, 800
SSE | ×5 |
| Stream3-Fusion | 10 | W/16 × H/16 | 1×1, 384
3×3, 384
SSE | ×1 | 1×1, 512
3×3, 512
SSE | ×1 | 1×1, 640
3×3, 640
SSE | ×1 | 1×1, 800
3×3, 800
SSE | ×1 |
| Downsampling | 11 | W/32 × H/32 | 1×1, 1280
3×3, 1280
stride 2 | ×1 | 1×1, 2048
3×3, 2048
stride 2 | ×1 | 1×1, 2560
3×3, 2560
stride 2 | ×1 | 1×1, 3200
3×3, 3200
stride 2 | ×1 |
| Final | 12 | 1×1 | average pool, 1000-d fc, softmax | | | | | | | |

Table A1: Specification of ParNet models used for ImageNet classification: ParNet-S, ParNet-M, ParNet-L, and ParNet-XL.

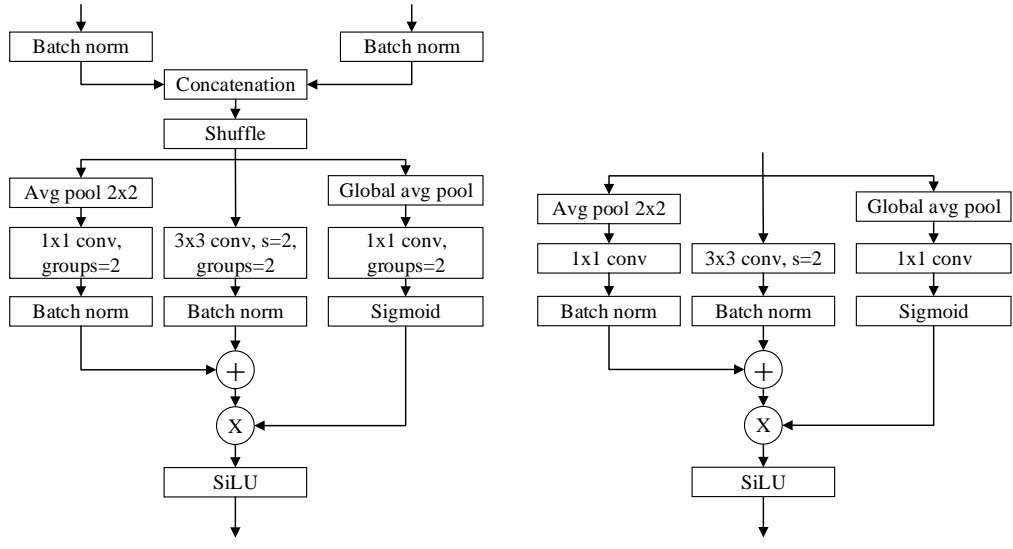

Figure A1: Schematic representation of the Fusion (left) and Downsampling (right) blocks used in ParNet.

