# OpenReview forum: "Non-deep Networks"
_ICLR.cc/2022/Conference — ICLR 2022 Submitted_

### Official Review · Reviewer_75Wm · 2021-10-31

**Correctness:** 2
**Technical Novelty And Significance:** 3
**Empirical Novelty And Significance:** 3
**Recommendation:** 5
**Confidence:** 5

**Main Review:**

Strength:
- empirical result that low-depth networks can be competitive with deeper models is significant. Confirming observations in prior works, the number of parameters and computational complexity of low-depth networks need to be much higher to match the accuracy of deeper networks.

Weaknesses:
- architecture complexity: the authors claim outperforming ResNet in efficiency, but do not mention that the architecture is far more complex. It is also more complex than RepVGG. This needs to be stressed in the introduction. This is separate from computational complexity.
- weak baseline: authors use Squeeze-and-Excitation blocks in their architecture which is known to improve ResNet performance, yet compare to ResNet without SE layers (section 4, tables 1-2)
- missing baseline: the authors introduce parallel streams in their network, which might be working as an ensemble, known to be improving classification performance on the tested datasets. An important baseline then is treating the stream subnetworks as separate models applied on different resolutions, which is simpler than the proposed architecture and would not suffer from communication overhead. A baseline with ResNet and ParNet ensembles needs to be added.
- statistical significance: test errors on CIFAR and ImageNet have high variance, and the authors show results of a single run for each experiment. A common practice to reduce variance is to train multiple networks and report mean and standard deviation for each experiment.
- SiLU: the authors replace ReLU with SiLU motivated by the limited representational power of ReLU. However, SiLU was introduced to improve training of deep networks, which is not the case in the proposed architecture. Moreover, in the ablation study (table 7) it does not seem to bring a significant improvement over ReLU, there is also no statistical significance analysis of this result. Using SiLU seems like a needless complication of the proposed architecture.

Arguable weakness:
- definition of depth: the approach takes advantage of the vague definition of depth in neural networks. In prior works, depth is defined as a number of layers in a single stream network. By introducing multiple parallel streams the authors effectively take a deep network and move layers around, so another, perhaps more fair, definition of depth could be counting the total number of layers in all substreams.

Notes to authors:
- table 3 is misleading because it does not show the number of GPUs used in the last row. It is also not clear if it is fused or not.

**Summary Of The Paper:**

The paper proposes a new architecture of a convolutional neural network for image classification. The authors are motivated by inference efficiency and showing that networks with about a dozen of layers can be competitive in classification accuracy with 50 layers and more. So instead of growing depth, they propose to grow width and have multiple subnetworks of the same depth within one model. The proposed architecture is evaluated on CIFAR, ImageNet and COCO classification and detection tasks, and shows that networks with 12 layers can be competitive with deeper counterparts.

**Summary Of The Review:**

The paper shows a significant result that a network with 12 layers can be competitive with deeper networks on well established image classification benchmarks. However, the architecture is more complex than ResNet or RepVGG, and the empirical evaluation is unsatisfactory due to weak and missing baselines, and the lack of statistical significance analysis. I hesitate between reject and weak reject.

---

> ### Author Response · Authors · 2021-11-24
> **Response to Reviewer 75Wm**
>
> Thank you so much feedback and suggestions. Following we have address your concerns:
>
> **Concern:** "architecture complexity: the authors claim outperforming ResNet in efficiency, but do not mention that the architecture is far more complex. It is also more complex than RepVGG. This needs to be stressed in the introduction. This is separate from computational complexity."
>
> **Response:** Exploring simple architectures is an exciting direction for research. Further investigation is required to build simpler non-deep networks, but this is not the focus of our work. We believe our work will enable progress in this direction by demonstrating the existence of high-performing non-deep networks.
>
> **Concern:** "weak baseline: authors use Squeeze-and-Excitation blocks in their architecture which is known to improve ResNet performance, yet compare to ResNet without SE layers (section 4, tables 1-2)"
>
> **Response:** "We compared with ResNets that use the Squeeze-and-Excitation layers in Table 8. For the same depth and similar parameter count, the RepVGG block performs better than the ResNet block."
>
> **Concern:** "missing baseline: the authors introduce parallel streams in their network, which might be working as an ensemble, known to be improving classification performance on the tested datasets. An important baseline then is treating the stream subnetworks as separate models applied on different resolutions, which is simpler than the proposed architecture and would not suffer from communication overhead. A baseline with ResNet and ParNet ensembles needs to be added."
>
> **Response:** "Thanks for the suggestion. We compared ParNet with ensembles for the same resolution in Table 9. Unfortunately, we could not complete this experiment now because of computational constraints."
>
> **Concern:** "statistical significance: test errors on CIFAR and ImageNet have high variance, and the authors show results of a single run for each experiment. A common practice to reduce variance is to train multiple networks and report mean and standard deviation for each experiment."
>
> **Response:** "We trained some representative models and did not see much fluctuation in accuracy. Specifically, the standard deviation in accuracy for the larger ParNet model on CIFAR10 and CIFAR100 was 0.13% and 0.19% respectively. Similarly, the standard deviation for ParNet-M on Imagenet was 0.1%"
>
> **Concern:** "SiLU: the authors replace ReLU with SiLU motivated by the limited representational power of ReLU. However, SiLU was introduced to improve training of deep networks, which is not the case in the proposed architecture. Moreover, in the ablation study (table 7) it does not seem to bring a significant improvement over ReLU, there is also no statistical significance analysis of this result. Using SiLU seems like a needless complication of the proposed architecture."
>
> **Response:** Thanks for the great suggestion. Removing SiLU can make our model faster and more useful for low end devices without losing much accuracy. We will do so in an updated version.
>
> **Concern:** "definition of depth: the approach takes advantage of the vague definition of depth in neural networks. In prior works, depth is defined as a number of layers in a single stream network. By introducing multiple parallel streams the authors effectively take a deep network and move layers around, so another, perhaps more fair, definition of depth could be counting the total number of layers in all substreams."
>
> **Response:** We disagree that our definition of depth is not fair. Depth is the minimum number of layers that must be run sequentially to get the output from input.
>
> **Concern:** "table 3 is misleading because it does not show the number of GPUs used in the last row. It is also not clear if it is fused or not."
>
> **Response:** Thanks for pointing out the confusion. We will update table 3 to clarify it further. Specifically, row 3 uses 3 GPUs and the fused model.

---

### Official Review · Reviewer_pTPq · 2021-11-02

**Correctness:** 3
**Technical Novelty And Significance:** 3
**Empirical Novelty And Significance:** 3
**Recommendation:** 5
**Confidence:** 4

**Main Review:**

Pros)
+ This paper is easy to follow.
+ The concept of branch-parallelism in a model looks promising.

Cons)
- I am agreeing that the network depth is a crucial design element towards model efficiency but the computational costs in including the number of parameters, FLOPs, and memory footprints seem to be overlooked in this paper: the proposed model has a larger # of parameters and FLOPs, so the overall computational costs are bigger than the baseline models.
- The architectural design with the proposed building block in Figure 2 is presented without providing any intuitions or design philosophy.
- The comparisons with the baselines in model-parallelism is not sufficient

Some comments and questions)
- The model-parallelism can be done with a single-branch model in training or inferring with multiple images (i.e., with a larger mini-batch size), then the speed gain from the branch-parallelism scheme may vanish. Please clarify a multi-branch architecture also has an advantage over the widened networks such as WideResNet in this case.

- The performance comparison should be done fairly. For example in Table 2, ParNet-L is compared with much smaller networks such as R34 and R50 in terms of the computational budgets, so this is not fair. The models with widening the width while fixing depth (e.g., Wide ResNet) can achieve much better accuracy barely increase the latency.

- Memory consumption would be a matter for the proposed architecture. Please specify the memory footprints when training (or inference) with the fixed batchsize.

- ResNets can also be leveraged fusing methods including skip connections and BNs. Did the authors compare the latency with these fused ResNets with the proposed ParNets in Table 2?

- Why vanilla SE-block is not suitable for the proposed model? Why Rep-VGG block has been adopted?

- Please specify why ParNets cannot outperform DenseNet-100 even using more parameters on the CIFAR datasets in Table 6. I am just wondering whether there is a different earning behavior of a shallow network on a particular dataset.

**Summary Of The Paper:**

This paper introduces a way of designing a network fixing the depth yet involving more branches. The authors try to support the claim which is that low-depth networks can achieve a good result by providing experiments on the ImageNet and the CIFAR datasets

**Summary Of The Review:**

- See the main review

---

> ### Author Response · Authors · 2021-11-24
> **Response to Reviewer pTPq (Part 1/2)**
>
> Thank you so much feedback and suggestions. Following we have address your concerns:
>
> **Concern:** "I am agreeing that the network depth is a crucial design element towards model efficiency but the computational costs in including the number of parameters, FLOPs, and memory footprints seem to be overlooked in this paper: the proposed model has a larger # of parameters and FLOPs, so the overall computational costs are bigger than the baseline models."
>
> **Response:** We acknowledge that ParNet consumes more memory and parameters than deeper counterparts.  But, as explained in the overall response, the trend in hardware accelerators suggests that tomorrow’s processors will have more cores and limited processor frequency. Hence, non-deep networks with more parallelism will benefit in spite of having more parameters and larger FLOPS. Further, there exists a large body of complementary techniques like pruning and distillation to build low parameter count and memory models.
>
> **Concern:** "The architectural design with the proposed building block in Figure 2 is presented without providing any intuitions or design philosophy."
>
> **Response:** The motivating philosophy of our architectural design is to build the lowest depth network. So we move the sequential processing units into parallel units. Surprisingly, even with these parallel units, the network is able to achieve high performance on large-scale benchmarks like ImageNet.
>
> **Concern:** "The comparisons with the baselines in model-parallelism is not sufficient. The model-parallelism can be done with a single-branch model in training or inferring with multiple images (i.e., with a larger mini-batch size), then the speed gain from the branch-parallelism scheme may vanish. Please clarify a multi-branch architecture also has an advantage over the widened networks such as WideResNet in this case."
>
> **Response:** We evaluate the latency of the models where samples are coming one at a time, like in robotics and autonomous driving. The batch size in all evaluations is 1. It is non-trivial to parallelize single-branch models across GPUs for batch size 1. In Table 8, we find multiple branches perform better than widened networks under the same parameter budget.
>
> **Concern:** "The performance comparison should be done fairly. For example in Table 2, ParNet-L is compared with much smaller networks such as R34 and R50 in terms of the computational budgets, so this is not fair. The models with widening the width while fixing depth (e.g., Wide ResNet) can achieve much better accuracy barely increase the latency."
>
> **Response:** In Table 2, we compared models with similar accuracy and speed. As suggested by the reviewer, we will also add numbers for ParNet-S and ParNet-M. Specifically, ParNet-S has an Top1 accuracy of 75.19% with speed of 280 samples/s , 19.2 millions parameters and 9.7 billions flops ; and ParNet-M has an Top1 accuracy of 76.60% with speed of 265 samples/s, 35.6 millions parameters and 17.2 billions flops.
>
> In Table 8, we compare with wide variants of ResNet and find that RepVGG-SSE blocks perform better.
>
> **Concern:** "Memory consumption would be a matter for the proposed architecture. Please specify the memory footprints when training (or inference) with the fixed batchsize."
>
> **Response:** Thanks for the suggestion. We now measure the memory consumption by comparing the largest batchsize that could fit in memory while inference We find that ParNet-L can use 1.25x times more memory than ResNet50.
>
> **Concern:**  "ResNets can also be leveraged fusing methods including skip connections and BNs. Did the authors compare the latency with these fused ResNets with the proposed ParNets in Table 2?"
>
> **Response:**  It is not possible to fuse the skip connection in ResNet as there is a non-linearity between the two convolution layers. However, as suggested, the BN in ResNet can be fused with a Convolution Layer. After doing this the speed of ResNets is increased but ParNet is still competitive. Specifically, ParNet-L achieves a top-1 accuracy of 77.66% and speed of 250 samples/s while ResNet-50 achieves a top-1 accuracy of 77.5% and speed of 240 samples/s.
>
> **Concern:**  "Why vanilla SE-block is not suitable for the proposed model? Why Rep-VGG block has been adopted?"
>
> **Response:** Vanilla SE-block increases depth of each block by 2 as it is applied sequentially after the convolution layer. Since we wanted to build a model with lower depth, we used the Skip SE variant which does not increase the depth of the model but provides the advantage of an SE block.

---

> > ### Author Response · Authors · 2021-11-24
> > **Response to Reviewer pTPq (Part 2/2)**
> >
> > **Concern:** "Please specify why ParNets cannot outperform DenseNet-100 even using more parameters on the CIFAR datasets in Table 6. I am just wondering whether there is a different earning behavior of a shallow network on a particular dataset."
> >
> > **Response:** A possible explanation for this is that deeper networks are more parameter efficient. Further, even among deep networks, it is empirically observed that DenseNet is more parameter efficient. Also, as suggested by the reviewer, parameter efficiency of shallow vs deep networks could depend on the property of the dataset as well. This is an exciting direction to explore, however, it is not the focus of our work.

---

### Official Review · Reviewer_Z9wZ · 2021-11-03

**Correctness:** 4
**Technical Novelty And Significance:** 3
**Empirical Novelty And Significance:** 3
**Recommendation:** 5
**Confidence:** 5

**Main Review:**

I have four considerations:
- (a) Table 2 only shows the details for ParNet-L and -XL. It could be better to show the full details and comparisons of ParNet-S, -M, -L, and -XL.
- (b) What are the details of speed testing in Table 2? Do the ResNet and ParNet use the same setting for speed benchmark? If ParNet uses multiple GPUs in parallel to benchmark the inference speed, e.g., 2 GPUs with 128 global batch size, does the ResNet uses 2 GPUs within the global batch size of 128 as well? It could be better to add more words in the third paragraph on Page 6 to show how to conduct the experiments.
- (c) ParNet develops the RepVGG-SSE based on the work of RepVGG, CVPR 2021. But there is no comparison with this high-related work. It could be better to compare with RepVGG in Table 1 and Table 2.
- (d) The paper argues that one of the advantages is parallel. But according to Table 2 and RepVGG's Table 4, ParNet has more parameters and larger FLOPs than the competitors. More parameters mean that the model will cost more memory during training and inference. This fact fades the significance of ParNet. It could be better to discuss this and highlight why parallelism is more important than the other two aspects for selling the work.

**Summary Of The Paper:**

- **Motivation**. The paper argues that deep networks have several limitations
  - (a) deep nets have a higher latency;
  - (b) deep nets are hard to parallelize;
  - (c) deep nets are not suitable for applications.


- **Method**.
Motivated by these observations, the paper aims to fill the performance gap between shallow networks and deep networks. The paper proposed a 12-layer shallow model framework, which contains proposed RepVGG-SSE blocks, fusion modules for multi-scale processing, and parallel streams.


- **Experiments**.
The proposed model, ParNet, is verified on CIFAR-10, CIFAR-100, and ImageNet for classification, MS-COCO for detection.

**Summary Of The Review:**

Please erase the above concerns.

---

> ### Author Response · Authors · 2021-11-24
> **Response to Reviewer Z9wZ**
>
> Thank you so much feedback and suggestions. Following we have address your concerns:
>
> **Concern:** "Table 2 only shows the details for ParNet-L and -XL. It could be better to show the full details and comparisons of ParNet-S, -M, -L, and -XL"
>
> **Response:** Thanks for the suggestion. We will do so in an updated version of the paper. Specifically, ParNet-S has an Top1 accuracy of 75.19% with speed of 280 samples/s , 19.2 millions parameters and 9.7 billions flops ; and ParNet-M has an Top1 accuracy of 76.60% with speed of 265 samples/s, 35.6 millions parameters and 17.2 billions flops.
>
> **Concern:** "What are the details of speed testing in Table 2? Do the ResNet and ParNet use the same setting for speed benchmark? If ParNet uses multiple GPUs in parallel to benchmark the inference speed, e.g., 2 GPUs with 128 global batch size, does the ResNet uses 2 GPUs within the global batch size of 128 as well? It could be better to add more words in the third paragraph on Page 6 to show how to conduct the experiments."
>
> **Response:** Thanks for pointing out the confusion. We evaluate the latency of the models where samples are coming one at a time like in robotics and autonomous driving. Hence, the batch size in all evaluations is 1. We will clarify it in the paper.
>
> **Concern:** "ParNet develops the RepVGG-SSE based on the work of RepVGG, CVPR 2021. But there is no comparison with this high-related work. It could be better to compare with RepVGG in Table 1 and Table 2"
>
> **Response:** Thanks for the great suggestion. We missed this comparison. But we have  this comparison now and find that for the same accuracy, RepVGG is 1.3 times faster than ParNet for large model (ParNet-XL vs RepVGG_b2g4) and by 1.5 times faster for medium-sized model (ParNet-L vs RepVGG_b1g4). We will add this comparison in the paper.
>
> RepVGG, although being 2.5 times deeper than ParNet, is faster as it consists of only optimized layers like 3X3 convolution and ReLU. However, as explained in the overall response, our contribution is in answering the scientific question of whether non-deep networks can achieve high performance on large-scale benchmarks. Further, non-deep networks are promising considering future hardware.
>
> **Concern:** "The paper argues that one of the advantages is parallel. But according to Table 2 and RepVGG's Table 4, ParNet has more parameters and larger FLOPs than the competitors. More parameters mean that the model will cost more memory during training and inference. This fact fades the significance of ParNet. It could be better to discuss this and highlight why parallelism is more important than the other two aspects for selling the work."
>
> **Response:** We acknowledge that ParNet consumes more memory and parameters than deeper counterparts. But, as explained in the overall response, the trend in hardware accelerators suggests that tomorrow’s processors will have more cores and limited processor frequency. Hence, non-deep networks with more parallelism will benefit in spite of having more parameters and larger FLOPS.

---

### Official Review · Reviewer_wcwX · 2021-11-04

**Correctness:** 3
**Technical Novelty And Significance:** 2
**Empirical Novelty And Significance:** 3
**Recommendation:** 5
**Confidence:** 5

**Main Review:**

1) My major concern is that the authors argues contribution of parallel subnetworks, while I think the traditional RepVGG-wise blocks adopted in ParNet contributes significantly more to prediction accuracy. Even if ParNet adopts 1 branch only, ParNet's accuracy can still achieve 75% more, as "ParNet-M-OneStream" shown in Table 8. Comparison of Table 7 and Table 10 shows that the RepVGG blocks is significantly more important than adopted number of branches on improving accuracy.

2) As shown in Table 2, ParNet-L/ParNet-XL (12-depth) indeed has fewer depth compared to ResNet50 (50-depth). However, ParNet-L (54.9M)/ParNet-XL (85.0M) has significantly more parameters compared to ResNet50 (25.6M). A potential problem is that if ParNet runs on low-power computing hardware with limited cuda cores, I am not sure whether ParNet still has the advantage on speed compared to ResNet50.

3) The paper proposes a new block RepVGG-SSE by introducing SSE into traditional RepVGG, while ParNet's accuracy improvement from SSE is only 1.53%, as shown in Table 7. Therefore, I think contribution of RepVGG SSE is not sufficient enough.

**Summary Of The Paper:**

The paper proposes to manually design a new 12-depth CNN architecture ParNet based on parallel subnetworks instead of traditionally deeply stacked blocks. Experiments show that ParNet is the first CNN achieving over 80% accuracy on ImageNet with 12 depth only. ParNet also achieves a competitive AP of 48% on MS-COCO for object detection.

**Summary Of The Review:**

The authors argues contribution of parallel subnetworks, while the experiments show that the traditional RepVGG-wise blocks adopted in ParNet contributes significantly more to prediction accuracy. Therefor, my rating is "5: marginally below the acceptance threshold".

---

> ### Author Response · Authors · 2021-11-24
> **Response to Reviewer wcwX**
>
> Thank you so much feedback and suggestions. Following we have address your concerns:
>
> **Concern:** "My major concern is that the authors argues contribution of parallel subnetworks, while I think the traditional RepVGG-wise blocks adopted in ParNet contributes significantly more to prediction accuracy. Even if ParNet adopts 1 branch only, ParNet's accuracy can still achieve 75% more, as "ParNet-M-OneStream" shown in Table 8. Comparison of Table 7 and Table 10 shows that the RepVGG blocks is significantly more important than adopted number of branches on improving accuracy."
>
> **Response:** We agree that the RepVGG-wise blocks adopted in ParNet are significant in our network. However, our contribution is in showing that RepVGG blocks, along with parallel sub-structures, can be used to build high-performing non-deep networks. As explained in the overall response, our novel contribution is the result, not any particular design choice. We will update our introduction to clarify this.
>
> **Concern:**  "As shown in Table 2, ParNet-L/ParNet-XL (12-depth) indeed has fewer depth compared to ResNet50 (50-depth). However, ParNet-L (54.9M)/ParNet-XL (85.0M) has significantly more parameters compared to ResNet50 (25.6M). A potential problem is that if ParNet runs on low-power computing hardware with limited cuda cores, I am not sure whether ParNet still has the advantage on speed compared to ResNet50"
>
> **Response:** We agree that the higher parameter count in ParNet makes it not suitable for cases like low-power computing hardware. But as explained in the overall response, non-deep networks are suitable for future hardware with more cores and limited processor frequency. Further, there exists an exciting body of complementary techniques like pruning and distillation to reduce the parameter count and memory of a model.
>
> **Concern:** "The paper proposes a new block RepVGG-SSE by introducing SSE into traditional RepVGG, while ParNet's accuracy improvement from SSE is only 1.53%, as shown in Table 7. Therefore, I think contribution of RepVGG SSE is not sufficient enough."
>
> **Response:** SSE helps in improving performance while not increasing depth. Since our objective is to build best performing non-deep networks, a 1.53% improvement on a large scale dataset like ImageNet is significant.

---

### Author Response · Authors · 2021-11-24
**Overall Response**

We thank the reviewers for their feedback and help in improving our work. We are excited that the reviewers found our results to be significant and our idea promising.

However, we believe that our paper is being evaluated as  a conventional neural network architecture design paper. Hence concerns have been raised about: first, novelty of the building block and second, compute (parameter count, flops, memory). We acknowledge that these  could be valid criteria for many works,  but it is not suitable for our work as it does not take into account our primary contribution.

Our work is motivated by the important scientific question “is it possible to build high-performing non-deep neural networks?” We show that it is possible to do so by adapting tools present in the literature. Although the tools are present in the literature, the result about high-performing non-deep networks is not. This result is our novel contribution.

For the second concern regarding compute, we acknowledge that currently non-deep networks are not a replacement for their deep counterparts in low-compute settings and further investigation is required in this direction. However, as explained in the paper, the trend in hardware accelerators (like GPUs) suggests that processor frequency is getting limited while the number of cores and memory is rising. This is favourable for non-deep networks as they depend less on processor frequency because of less depth; and more on the number of cores because of wide parallel processing. Considering this trend, non-deep networks are promising considering tomorrow’s hardware, even though they consume more compute (flops and parameters).

We will update our introduction and abstract to further clarify this. Below we have addressed the individual concerns of the reviewers.

---

### Decision · Program_Chairs · 2022-01-20

**Decision:**

Reject

**Comment:**

This paper shows the possibility to design a relatively shallow architecture, ParNet, based on parallel subnetworks, instead of traditionally deeply stacked blocks. During discussions, the reviewers pointed out two important concerns: (1) the current design heavily hinges on the recently proposed RepVGG block, whose comparison was even missed in the original submission (later added in rebuttal); (2) comparing ParNet with RepVGG, there seems no performance advantage. Although RepVGG is 2.5 times deeper than ParNet, it is still faster due to highly optimized layers.

The authors mainly argued that their contribution is to answer the scientific question “is it possible to build high-performing non-deep neural networks?” While this is indeed an interesting question, AC feels: (1) it is perhaps unfair for this paper to claim as the first work proving the feasibility. WideResNet provided similar insight much earlier, among others; (2) the presented results, with tools being not novel, are pre-mature as they display no real appeal of using ParNet, in any aspect. Probing a new question is of course valuable, but presenting an immature and novelty-lacking answer shouldn't automatically grant publication.

In sum, the reviewers were unanimously UN-convinced by this paper's value, nor was the AC. The authors are suggested to very seriously take into account reviewers' suggestions to make improvements, before submitting their work to the next venue.